# A Site-Ordered Quadruple Perovskites, RMn_3_Ni_2_Mn_2_O_12_ with R = Bi, Ce, and Ho, with Different Degrees of B Site Ordering

**DOI:** 10.3390/molecules30081749

**Published:** 2025-04-14

**Authors:** Alexei A. Belik

**Affiliations:** Research Center for Materials Nanoarchitectonics (MANA), National Institute for Materials Science (NIMS), Namiki 1-1, Tsukuba 305-0044, Ibaraki, Japan; alexei.belik@nims.go.jp

**Keywords:** A site-ordered quadruple perovskites, B site double ordering, crystal structures, structural disorder, magnetic properties

## Abstract

A site-ordered quadruple perovskites, AA′_3_B_4_O_12_, can have 3*d* transition metals at A′ and B sites, and show complex magnetic interactions and behavior. Additional complexity appears when B site-ordered arrangements are realized in AA′_3_B_2_B′_2_O_12_. In this work, A site-ordered quadruple perovskites, RMn_3_Ni_2_Mn_2_O_12_ with R = Bi, Ce, and Ho, were prepared by a high-pressure, high-temperature method at about 6 GPa and about 1500 K. The R = Bi and Ce samples were found to crystallize in space group *Im*-3 with a disordered distribution of Ni^2+^ and Mn^4+^ cations in one B site. On the other hand, the R = Ho sample crystallized in space group *Pn*-3 and showed partial ordering of Ni^2+^ and Mn^4+^ cations between two B sites. The structural data (and bond valence sums) suggest that cerium has the oxidation state +3, which is unusual for such perovskites. Magnetic properties were investigated by magnetic susceptibility and specific heat measurements, which showed the presence of one magnetic transition near 36 K for R = Bi; there was evidence for the presence of two magnetic transitions near 27 K and 33 K for R = Ce, and near 10 K and 36 K for R = Ho. Curie–Weiss parameters were estimated for all samples from high-temperature magnetic measurements up to 750 K. The total effective magnetic moment for R = Ce also suggests the presence of Ce^3+^. A magnetic field of 90 kOe had the largest effect on the specific heat of the R = Ho sample, and almost no effects on the specific heat of the R = Bi sample.

## 1. Introduction

Simple perovskite oxides, ABO_3_, usually have non-magnetic A cations (such as, Ca^2+^, Sr^2+^, Ba^2+^, Pb^2+^, La^3+^, Y^3+^, and Bi^3+^) or magnetic rare-earth cations [1,2,3,4,5]. Therefore, the magnetism of ABO_3_ perovskites is primarily driven by B cations (if they are magnetic). Even if both A and B cations are magnetic, they behave separately at higher temperatures, and the interaction between A and B cations is weak. For example, RFeO_3_ perovskites have Fe^3+^ ordering at about 600–740 K, while R^3+^ ordering takes place below about 10 K [6].

On the other hand, A site-ordered quadruple perovskites, AA′_3_B_4_O_12_, have a small square-planar A′ site, which usually accommodates 3*d* transition metals [7,8,9,10,11]. Therefore, such perovskites can show complex magnetic interactions and behavior and much stronger interactions between cations in the A′ and B sites. Additional complexity can appear when different cations are located at the B sites. Depending on the size and charge difference [2], B site-ordered arrangements can be realized, leading to the general formula for such perovskites being AA′_3_B_2_B′_2_O_12_ [12,13,14,15,16,17,18,19,20,21,22,23,24,25,26,27,28]. High degrees of B site cation orders are usually realized when the charge difference between B and B′ cations is higher than 3, or when there is a large difference in the electronic properties between B and B′ cations [2]. When the charge difference between B and B′ cations is equal to 2, synthesis conditions start to play a very important role in the degree of cation order [2].

Special attention has been given in the literature to perovskite-related compounds with B = Ni^2+^ and B′ = Mn^4+^ [29,30,31,32,33,34,35,36,37,38,39,40,41,42,43,44,45,46], because such a combination is expected to give ferromagnetic (FM) interactions according to the Goodenough–Kanamori rules [47]. The charge difference between Ni^2+^ and Mn^4+^ is 2; therefore, it is often challenging to obtain fully ordered samples.

Simple R_2_NiMnO_6_ perovskites can be prepared at ambient pressure from R = La to R = Lu. The degree of Ni^2+^ and Mn^4+^ ordering has significant effects on the FM properties of R_2_NiMnO_6_ [35,36,37,38] and on the saturation magnetization values. In nearly ordered samples, *T*_C_ (*T*_C_ is an FM Curie temperature) decreases almost linearly from 280 K for R = La [30,36] to 40 K for to R = Lu [12,44,45]. Such changes in *T*_C_ correlate with changes in the Ni–O–Mn bond angles. Further reduction in the Ni–O–Mn bond angle results in dramatic changes in magnetic interactions and properties. For example, Sc_2_NiMnO_6_, even with full ordering of Ni^2+^ and Mn^4+^, shows two antiferromagnetic (AFM) transitions and a complex magnetodielectric response [41]. In_2_NiMnO_6_, also with full ordering of Ni^2+^ and Mn^4+^, demonstrates a complex incommensurate AFM structure below *T*_N_ = 26 K (*T*_N_ is an AFM Néel temperature) and spin-induced ferroelectric polarization at *T*_N_ [39,40]. As Lu_2_NiMnO_6_ appears to be located near a phase boundary between FM and AFM ground states, a moderate pressure can change Ni–O–Mn bond angles enough to induce a transition from an FM ground state to an incommensurate AFM ground state [44]. It should be noted that Sc_2_NiMnO_6_ and In_2_NiMnO_6_ can only be prepared using a high-pressure, high-temperature method.

Peculiarities of both the R_2_NiMnO_6_ and AA′_3_B_2_B′_2_O_12_ perovskite families are combined in the case of LaMn_3_Ni_2_Mn_2_O_12_ [12], which can be prepared only using a high-pressure, high-temperature method. LaMn_3_Ni_2_Mn_2_O_12_ is located in a region of a phase diagram with unusual properties similar to those of Sc_2_NiMnO_6_ and In_2_NiMnO_6_. For example, two magnetic transitions were found in LaMn_3_Ni_2_Mn_2_O_12_ at 34 K and 46 K with original magnetic structures [12,13]. Such perovskites were recently extended to RMn_3_Ni_2_Mn_2_O_12_ with R = Nd, Sm, Gd, and Dy [48], which can also be prepared only using a high-pressure, high-temperature method.

In this work, we investigated RMn_3_Ni_2_Mn_2_O_12_ with R = Bi, Ce, and Ho. Bi-containing perovskites are exceptional members of any perovskite family [49,50,51,52,53] in the majority of cases, as a Bi^3+^ cation has the lone electron pair and promotes polar distortions. Ce usually has the oxidation state of +4; therefore, Ce-containing members of any perovskite family may differ from other members of the same series with R^3+^. Perovskites with different oxidation states of Ce are reported in the literature [54,55,56,57,58,59,60,61,62]. Ho^3+^ is the smallest cation for which RMn_3_Ni_2_Mn_2_O_12_ perovskites have been prepared so far.

## 2. Results and Discussion

All prepared RMn_3_Ni_2_Mn_2_O_12_ samples with R = Bi, Ce, and Ho had a cubic crystal structure, as all reflections belonging to RMn_3_Ni_2_Mn_2_O_12_ could be indexed in a cubic system. The samples also contained different impurities depending on R: the R = Bi sample had Bi_2_O_2_CO_3_, NiO, NiMnO_3_, and GdFeO_3_-type impurities, the R = Ce had CeO_2_, NiO, and NiMnO_3_ impurities, and the R = Ho sample had HoMn_2_O_5_, NiO, and NiMnO_3_ impurities. All cubic reflections of the R = Bi and Ce samples could be indexed in space group *Im*-3, which is the parent (maximum) symmetry of the AA′_3_B_4_O_12_-type perovskites [7,8,9,10,11]. On the other hand, a (311) reflection was observed in the synchrotron XRPD data of the R = Ho sample (the inset of Figure 1a). This reflection is forbidden in space group *Im*-3, and suggests space group *Pn*-3, which is observed in cases of full or partial B site ordering of AA′_3_B_2_B′_2_O_12_-type perovskites [12,14,15,16,17,18,19].

The distribution of Ni^2+^ and Mn^4+^ cations between the B and B′ sites in the R = Ho sample was refined with the following constraints: (1) the full site occupation, *g*(Mn) + *g*(Ni) = 1, where g is the occupation factor, should be kept, and (2) the total chemical composition should be kept. The obtained distribution of Ni^2+^ and Mn^4+^ cations was close to 0.8Ni^2+^ + 0.2Mn^4+^ for the B site and 0.2Ni^2+^ + 0.8Mn^4+^ for the B′ site. These values suggest a significant degree of Ni^2+^ and Mn^4+^ ordering. We note that similar distributions of Ni^2+^ and Mn^4+^ cations were found in samples with R = Nd, Sm, Gd, and Dy [48] prepared under the same conditions.

The refinement of the occupation factors of the R sites (together with atomic displacement parameters (ADPs) of the R sites and all other refined parameters) gave the following values: *g*(Bi) = 1.0001(13), *g*(Ce) = 0.9710(15), and *g*(Ho) = 0.9929(14). These values were close to 1 within the sensitivity of the method, as such deviations could be absorbed by reasonable ADPs. Therefore, we fixed the occupation factor of the R sites at 1 in the final reported models. We emphasize that the ADP for Bi was enhanced. However, all attempts to split (to reduce the ADP of Bi) the Bi site from the ideal (0, 0, 0) position to different positions failed. Enhanced ADPs for Bi are often observed in such perovskites with a cubic structure, even in neutron diffraction data [52]. Enhanced ADPs for Bi are caused by the effect of the lone electron pair of Bi^3+^ cations and random displacements of Bi^3+^ cations.

The refined structural parameters of HoMn_3_Ni_2_Mn_2_O_12_ are summarized in Table 1, and those of RMn_3_Ni_2_Mn_2_O_12_ with R = Bi and Ce in Table 2. Experimental, calculated, and difference synchrotron X-ray powder diffraction patterns are shown in Figure 1 for HoMn_3_Ni_2_Mn_2_O_12_ and CeMn_3_Ni_2_Mn_2_O_12_. Appendix A gives experimental, calculated, and difference synchrotron X-ray powder diffraction patterns for BiMn_3_Ni_2_Mn_2_O_12_.

The bond valence sum (BVS) values [63,64] of the Mn_SQ_ site (where SQ stands for square-planar) were close to +3 for all three compounds (+2.94 for R = Bi, +2.97 for R = Ce, and +2.95 for R = Ho) suggesting that this site is occupied by Mn^3+^ cations. The BVS values of the Ni/Mn site (with *R*_0_ = 1.7035, which is the average value of *R*_0_(Ni^2+^) and *R*_0_(Mn^4+^) [63]) were +2.90 for R = Bi and +2.89 for R = Ce, again close to the average oxidation state of Ni^2+^ and Mn^4+^ cations. On the other hand, for R = Ho, the BVS value of the Ni_1_/Mn_1_ site (with the majority of Ni^2+^) was +2.28, and the BVS value of the Mn_2_/Ni_2_ site (with the majority of Mn^4+^) was +3.98, in agreement with the expected values. The BVS value of the Ni_1_/Mn_1_ site was somewhat higher than +2; however, similar values were observed in all other members of the RMn_3_Ni_2_Mn_2_O_12_ family with R = La [12], Nd, Sm, Gd, and Dy [48].

The BVS values of the R site were +2.71 for R = Bi, +3.04 for R = Ce [64], and +2.63 for R = Ho. Reduced BVS values of Bi^3+^ cations are often observed in Bi-containing A site-ordered quadruple perovskites [49,52], and could be caused by the influence of the lone electron pair of Bi^3+^ cations. The BVS value for Ho^3+^ suggests that Ho^3+^ cations are highly underbonded inside the A site. The severe underbonding could be a reason why RMn_3_Ni_2_Mn_2_O_12_ compounds become unstable with smaller R^3+^ cations [65], such as with R = Er-Lu [48], under the synthesis conditions used. In RMn_3_Ni_2_Mn_2_O_12_, the BVS values of R^3+^ cations changed from +3.40 for R = La [12] to +2.72 for R = Dy [48]. Therefore, the BVS value for R = Ho agrees with the general tendency in such perovskites. These BVS values also suggest that cations at the A site can be highly overbonded or highly underbonded, but the structure can still tolerate such stress. Stabilization of the structure was realized through the high-pressure, high-temperature preparation method. The BVS value for Ce suggests that Ce has an oxidation state of +3. Therefore, the reduction of Ce^4+^ (added as CeO_2_) took place during the synthesis of CeMn_3_Ni_2_Mn_2_O_12_, and CeMn_3_Ni_2_Mn_2_O_12_ is not an exceptional member of the R^3+^Mn_3_Ni_2_Mn_2_O_12_ series. The reduction of Ce^4+^ occurred through the oxidation of a part of Mn^3+^, as the high-pressure synthesis was performed in sealed capsules in closed environment.

The results of the specific heat measurements are given in Figure 2, Figure 3 and Figure 4. None of the samples showed sharp peaks on specific heat curves. Nevertheless, clear anomalies were observed that could be assigned to long-range magnetic orderings. BiMn_3_Ni_2_Mn_2_O_12_ showed a clear kink at 36 K (Figure 2), below which the *C*_p_/*T* values started decreasing sharply. High magnetic fields (up to 90 kOe) had almost no effects on the specific heat values of BiMn_3_Ni_2_Mn_2_O_12_, suggesting that the magnetic state was quite robust. CeMn_3_Ni_2_Mn_2_O_12_ showed a broadened kink at 33 K (Figure 3) and a second kink at 27 K, below which the *C*_p_/*T* values started decreasing. High magnetic fields (up to 90 kOe) had weak effects on the specific heat values of CeMn_3_Ni_2_Mn_2_O_12_, and moved the positions of the broad anomaly to slightly higher temperatures. HoMn_3_Ni_2_Mn_2_O_12_ showed two broadened peaks near 36 K and 10 K (Figure 4). High magnetic fields (up to 90 kOe) noticeably smeared the anomaly near 10 K, and moved the positions of the second broad anomaly to slightly lower temperatures. The specific heat data of HoMn_3_Ni_2_Mn_2_O_12_ were very close to the specific heat curves of DyMn_3_Ni_2_Mn_2_O_12_ [48].

The temperature dependence of the magnetic susceptibility of BiMn_3_Ni_2_Mn_2_O_12_ is shown in Figure 5a, and Figure 5b gives an *M* versus *H* curve at 5 K, 60 K, and 300 K. The divergence between the ZFC and FCC curves was observed below about 15 K at *H* = 10 kOe; at the same temperature, a broad maximum was observed on the ZFC curve. However, no clear anomalies were detected near 36 K, even on differential curves. Therefore, only specific heat data could clearly identify a magnetic phase transition temperature in BiMn_3_Ni_2_Mn_2_O_12_. The *M* versus *H* curves showed a very weak step-like hysteresis near the origin. The hysteresis near the origin probably originated from a small amount of ferrimagnetic NiMnO_3_ impurity (with *T*_C_ = 430 K [66]) with soft ferrimagnetic properties, as the hysteresis was observed even at 300 K. A slim extended hysteresis observed at 5 K (the inset of Figure 5b) originated from BiMn_3_Ni_2_Mn_2_O_12_.

The temperature dependence of the magnetic susceptibility of CeMn_3_Ni_2_Mn_2_O_12_ is shown in Figure 6a, and Figure 6b gives an *M* versus *H* curve at 5 K, 60 K, and 300 K. The divergence between the ZFC and FCC curves was observed below about 10 K at *H* = 10 kOe; at nearly the same temperature, a broad maximum was observed on the ZFC curve. The differential curves showed some anomalies; for example, the *dχT*/*dT* versus *T* curves at *H* = 100 Oe showed peaks at 27 K with shoulders at 33 K, in agreement with the specific heat data; on the other hand, at *H* = 10 kOe, only one broad anomaly was seen at 33 K. The *M* versus *H* curve at 5 K demonstrated a slim extended hysteresis. Above *T*_N_ (at 60 K and 300 K), a very weak, step-like hysteresis was observed near the origin, originating from ferrimagnetic NiMnO_3_ impurity [66]. Therefore, the extended hysteresis at 5 K should originate from the main phase.

The temperature dependence of the magnetic susceptibility of HoMn_3_Ni_2_Mn_2_O_12_ is shown in Figure 7a, and Figure 7b gives *M* versus *H* curves at 5 K and 60 K. In this case, kinks on the *χ* versus *T* curves were clearly observed at 36 K. The differential *dχ*/*dT* versus *T* curves at *H* = 100 Oe and 10 kOe showed sharp anomalies at 36 K and broader anomalies near 10 K, in agreement with the specific heat data. The *M* versus *H* curves showed almost no hysteresis. No detectable step-like hysteresis was observed near the origin at 60 K (above *T*_N_ of HoMn_3_Ni_2_Mn_2_O_12_) from NiMnO_3_ impurity, because of its very small amount. The *χ* versus *T* and *M* versus *H* curves of HoMn_3_Ni_2_Mn_2_O_12_ were very close to those of DyMn_3_Ni_2_Mn_2_O_12_ [48]. The ground states of Ho^3+^ and Dy^3+^ cations are, of course, different [67]. Nevertheless, these cations are quite close to each other, as they have close ionic radii (r_VIII_(Dy^3+^) = 1.027 Å versus r_VIII_(Ho^3+^) = 1.015 Å [65]), effective calculated magnetic moments (10.63*μ*_B_ for Dy^3+^ versus 10.60*μ*_B_ for Ho^3+^ [67]), and maximum ordered magnetic moments (of 10*μ*_B_). Therefore, it could be expected that HoMn_3_Ni_2_Mn_2_O_12_ and DyMn_3_Ni_2_Mn_2_O_12_ would have similar magnetic properties and ground states.

NiMnO_3_ is a ferrimagnetic material with a *T*_C_ of about 430 K [66]. Therefore, it gives noticeable contributions to the measured magnetic properties in small magnetic fields (such as 100 Oe (Appendix A)) below 400 K (Figure 8). As NiMnO_3_ is a soft ferrimagnetic material, its magnetization saturates at high magnetic fields. Therefore, the larger the measurement magnetic field, the smaller the relative contribution of NiMnO_3_. However, we found that even at 70 kOe (the maximum available magnetic field), contributions from NiMnO_3_ affected parameters of the Curie–Weiss fits below 400 K (Figure 8). Therefore, we performed high-temperature magnetic measurements up to 750 K. Above about 440 K, magnetic susceptibilities were independent of the measurement magnetic field, and reliable, intrinsic Curie–Weiss parameters could be obtained. Inverse magnetic susceptibilities followed the Curie–Weiss law at high temperatures. The Curie–Weiss fits were performed between 480 and 750 K (for the 70 kOe FCC curves), and the resultant parameters are summarized in Table 3. The experimental effective magnetic moments (*μ*_eff_) were very close to the calculated values (*μ*_calc_), supporting the charges of magnetic cations. In particular, the inclusion of the magnetic moment of Ce^3+^ [67] in the calculation gave a better agreement between *μ*_eff_ and *μ*_calc_; this fact gives indirect support for the +3 oxidation state of Ce, in addition to the structural data. We note that the Curie–Weiss temperatures (*θ*) were very small for the R = Ce and Ho samples when the fits were performed between 200 K and 380 K at 70 kOe (Figure 8).

RMn_3_Ni_2_Mn_2_O_12_ samples with R = Bi, Ce, and Ho were prepared under the same synthesis conditions (6 GPa and 1500 K). The same synthesis conditions were used for the preparation of RMn_3_Ni_2_Mn_2_O_12_ with R = Nd, Sm, Gd, and Dy [48]. These conditions produced partial ordering of Ni^2+^ and Mn^4+^ cations in the case of R = Nd-Ho. It was also found that an increase in the synthesis temperature to 1700 K produced samples with a disordered arrangement of Ni^2+^ and Mn^4+^ cations in the case of R = Nd and Sm [48]. Therefore, the disordered arrangement of Ni^2+^ and Mn^4+^ cations found in the R = Bi and Ce samples prepared at 1500 K could suggest that a (partial) ordering may be realized at lower synthesis temperatures. This possibility may be explored in the future. Note that larger amounts of impurities in the R = Ce sample (Table 2) could shift the real chemical composition of the main phase and contribute to the stabilization of the disordered structure.

## 3. Experimental

RMn_3_Ni_2_Mn_2_O_12_ samples with R = Bi, Ce, and Ho were prepared from stoichiometric mixtures of Bi_2_O_3_ (Rare Metallic Co., Tokyo, Japan, 99.9999%), CeO_2_ (Rare Metallic Co., Tokyo, Japan, 99.99%), Ho_2_O_3_ (Rare Metallic Co., Tokyo, Japan, 99.9%), Mn_2_O_3_ (Rare Metallic Co., Tokyo, Japan, 99.99%), MnO_2_ (Alfa Aesar, Ward Hill, MA, USA, 99.99%), and NiO (Rare Metallic Co., Tokyo, Japan, 99.9%). Single-phase Mn_2_O_3_ was prepared from a commercial MnO_2_ chemical (Rare Metallic Co., Tokyo, Japan, 99.99%) by annealing in air at 923 K for 24 h. The synthesis was performed at about 6 GPa and at about 1500 K for 2 h in sealed Au capsules, using a belt-type HP instrument. After annealing at 1500 K, the samples were cooled down to room temperature by turning off the heating current, and the pressure was slowly released.

X-ray powder diffraction (XRPD) data were collected at room temperature on a MiniFlex600 diffractometer (Rigaku, Tokyo, Japan) using CuKα radiation (2*θ* range of 8–100°, step width of 0.02°, and scan speed of 2°/min). Room-temperature synchrotron XRPD data were measured using the beamline BL02B2 [68,69] of SPring-8, Japan. Intensity data were collected between 1.00° and 83.02° at a 0.006° interval in 2*θ* with 100 s exposure time. A wavelength of *λ* = 0.4137875 Å was used for R = Bi (to reduce the absorption effect because of the presence of heavy Bi). A wavelength of *λ* = 0.6200666 Å was used for R = Ce and Ho. The samples were placed into open Lindemann glass capillary tubes (inner diameter: 0.2 mm), which were rotated during measurements. The Rietveld analysis of all XRPD data was performed using the *RIETAN-2000* program [70]. Normalized (because of different measurement times) background (obtained through measurements of (different) empty capillary tubes) was subtracted from the raw synchrotron XRPD data. Background subtraction usually gives larger profile agreement factors (*R*_wp_ and *R*_p_) because background intensities become very low. On the other hand, the background subtraction usually improves integrated agreement factors (*R*_I_ and *R*_F_) because better structure descriptions are achieved. The remaining background was fitted by 12-order Legendre polynomials. We note that NiMnO_3_ impurity was fitted using the ilmenite-type model (space group *R*-3) in the R = Bi and Ce samples, and using the simple corundum-type model (space group *R*-3*c*) in the R = Ho sample, because of its very small amount.

Magnetic measurements were performed on SQUID magnetometers (Quantum Design MPMS3, San Diego, CA, USA) between 2 and 400 K in applied fields of 100 Oe, 10 kOe, and 70 kOe, under both zero-field-cooled (ZFC) and field-cooled on cooling (FCC) conditions. High-temperature magnetic measurements were performed between 330 K and 750 K with the same magnetic fields using a high-temperature MPMS3 option, where dense pellets of the samples with polished flat surfaces were attached to a high-temperature stick with Zircar cement. Isothermal magnetization was measured at temperatures of 5 K, 60 K, and 300 K, between −70 and 70 kOe.

The specific heat, *C*_p_, was measured on cooling from 100 K to 2 K at different magnetic fields up to 90 kOe by a pulse relaxation method, using a commercial calorimeter (Quantum Design PPMS, San Diego, CA, USA). All magnetic and specific heat measurements were performed using pieces of dense pellets.

## 4. Conclusions

The synthesis of A site-ordered quadruple perovskites, RMn_3_Ni_2_Mn_2_O_12_ with R = Bi, Ce, and Ho, was performed by a high-pressure, high-temperature method at about 6 GPa and about 1500 K. Despite using the same synthesis conditions, the R = Bi and Ce samples crystallized in space group *Im*-3 with a disordered arrangement of Ni^2+^ and Mn^4+^ cations. On the other hand, the R = Ho sample crystallized in space group *Pn*-3 with partial ordering of Ni^2+^ and Mn^4+^ cations. By using a combination of magnetic susceptibility and specific heat measurements, one magnetic transition was detected in the R = Bi sample near 36 K. On the other hand, two magnetic transitions were found near 27 K and 33 K in the R = Ce sample, and near 10 K and 36 K in the R = Ho sample. Magnetic properties were affected by the presence of ferrimagnetic NiMnO_3_ impurity (with a *T*_C_ of about 430 K). Therefore, Curie–Weiss parameters were estimated for all samples from high-temperature magnetic measurements up to 750 K. The total effective magnetic moment for R = Ce and structural information, obtained from the state-of-the-art synchrotron X-ray powder diffraction, suggested the presence of Ce^3+^.

## Figures and Tables

**Figure 1 molecules-30-01749-f001:**
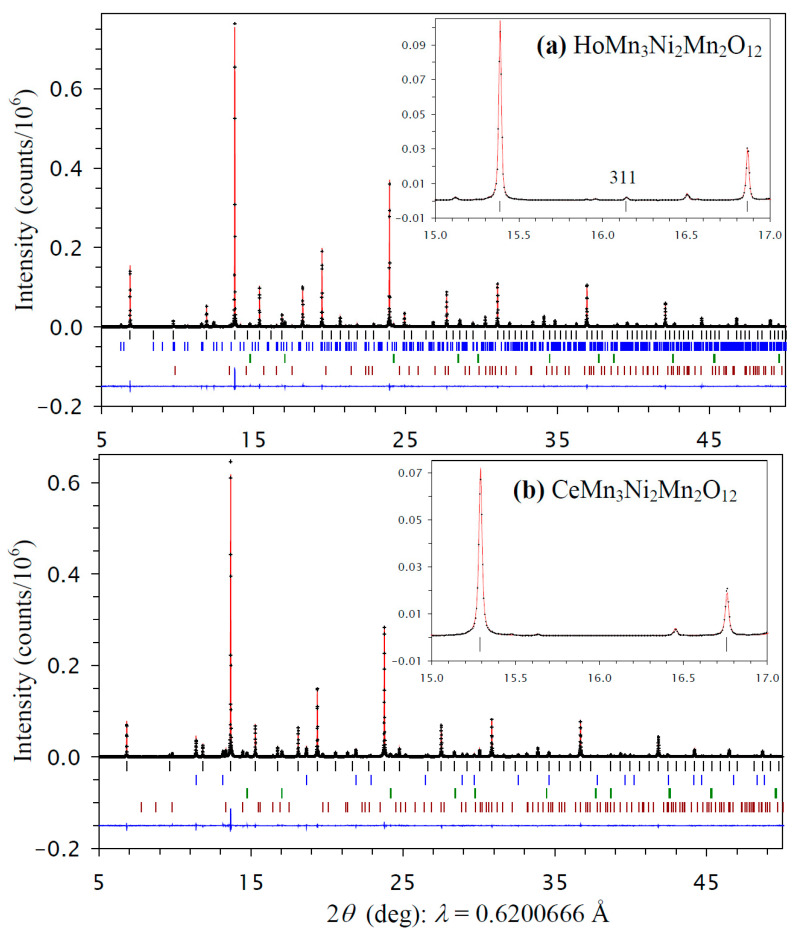
Fragments of experimental (black crosses), calculated (red line), and difference (blue line at the bottom) room-temperature synchrotron X-ray powder diffraction patterns of (**a**) HoMn_3_Ni_2_Mn_2_O_12_ (the *Pn*-3 modification) and (**b**) CeMn_3_Ni_2_Mn_2_O_12_ (the *Im*-3 modification) in a 2*θ* range of 5° and 50°. The tick marks show possible Bragg reflection positions for the main phase (black) and impurities (from top to bottom for HoMn_2_O_5_ (blue), NiO (green), and NiMnO_3_ (brown) for R = Ho and for CeO_2_ (blue), NiO (green), and NiMnO_3_ (brown) for R = Ce). Insets show magnified parts in a 2*θ* range of 15° and 17°, and emphasize the presence of the (311) reflection for R = Ho from the B site ordering and the absence of such a reflection for R = Ce.

**Figure 2 molecules-30-01749-f002:**
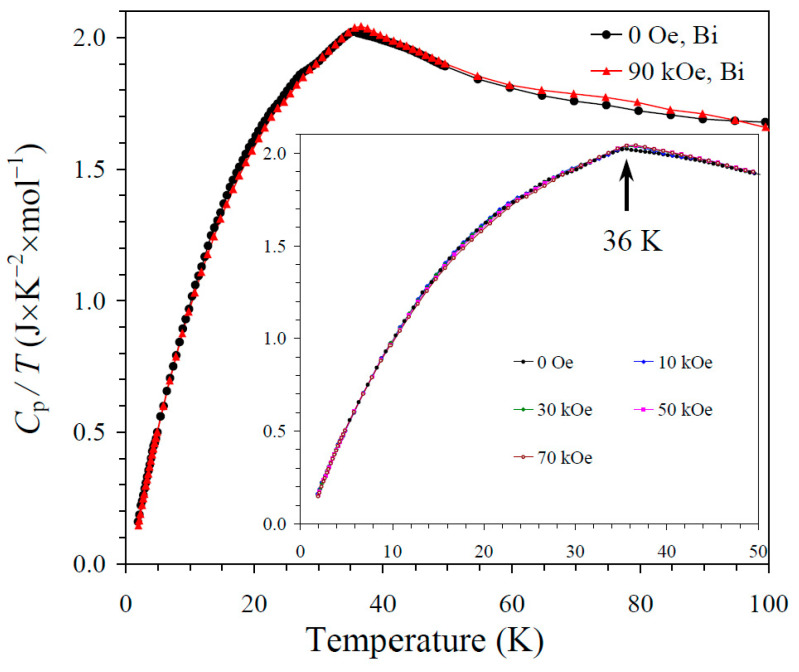
*C*_p_/*T* versus *T* curves of BiMn_3_Ni_2_Mn_2_O_12_ measured at *H* = 0 (black curves) and 90 kOe (red curves). The inset shows *C*_p_/*T* versus *T* curves at *H* = 0, 10, 30, 50, and 70 kOe. The arrow shows the magnetic transition temperature.

**Figure 3 molecules-30-01749-f003:**
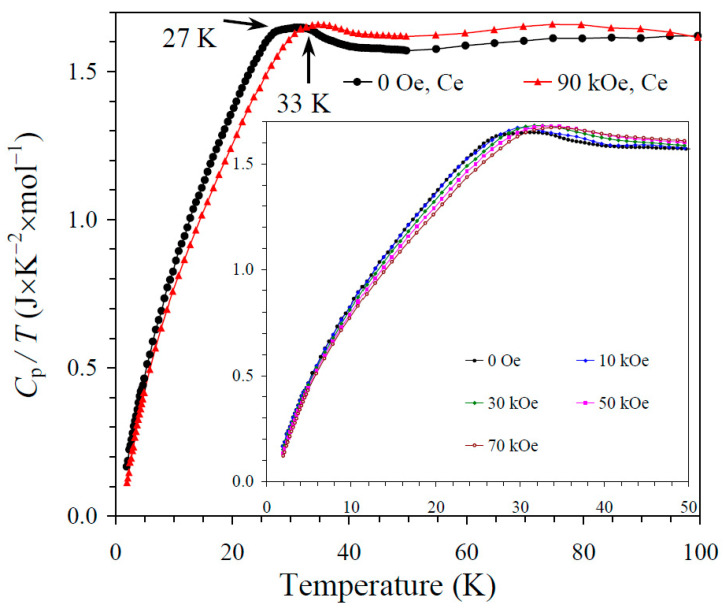
*C*_p_/*T* versus *T* curves of CeMn_3_Ni_2_Mn_2_O_12_ measured at *H* = 0 (black curves) and 90 kOe (red curves). The inset shows *C*_p_/*T* versus *T* curves at *H* = 0, 10, 30, 50, and 70 kOe. The arrows show magnetic transition temperatures.

**Figure 4 molecules-30-01749-f004:**
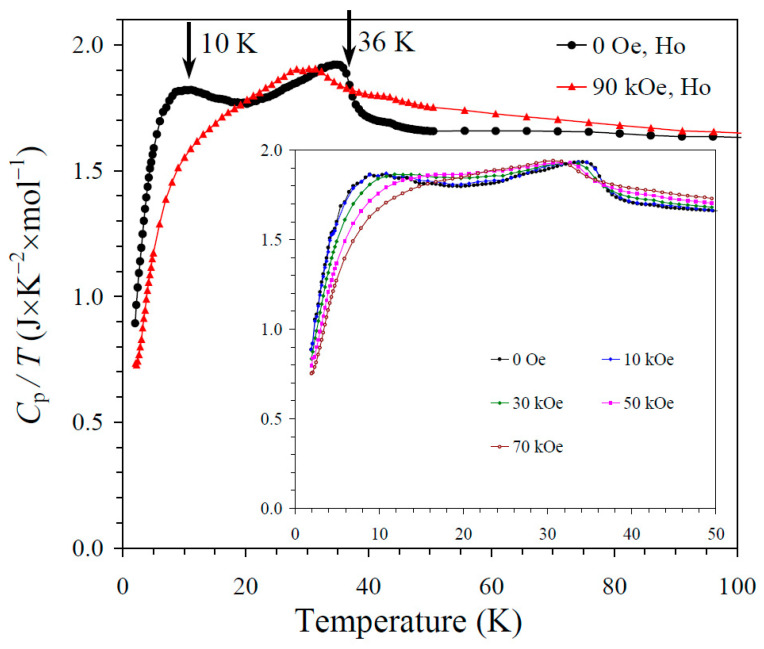
*C*_p_/*T* versus *T* curves of HoMn_3_Ni_2_Mn_2_O_12_ measured at *H* = 0 (black curves) and 90 kOe (red curves). The inset shows *C*_p_/*T* versus *T* curves at *H* = 0, 10, 30, 50, and 70 kOe. The arrows show magnetic transition temperatures.

**Figure 5 molecules-30-01749-f005:**
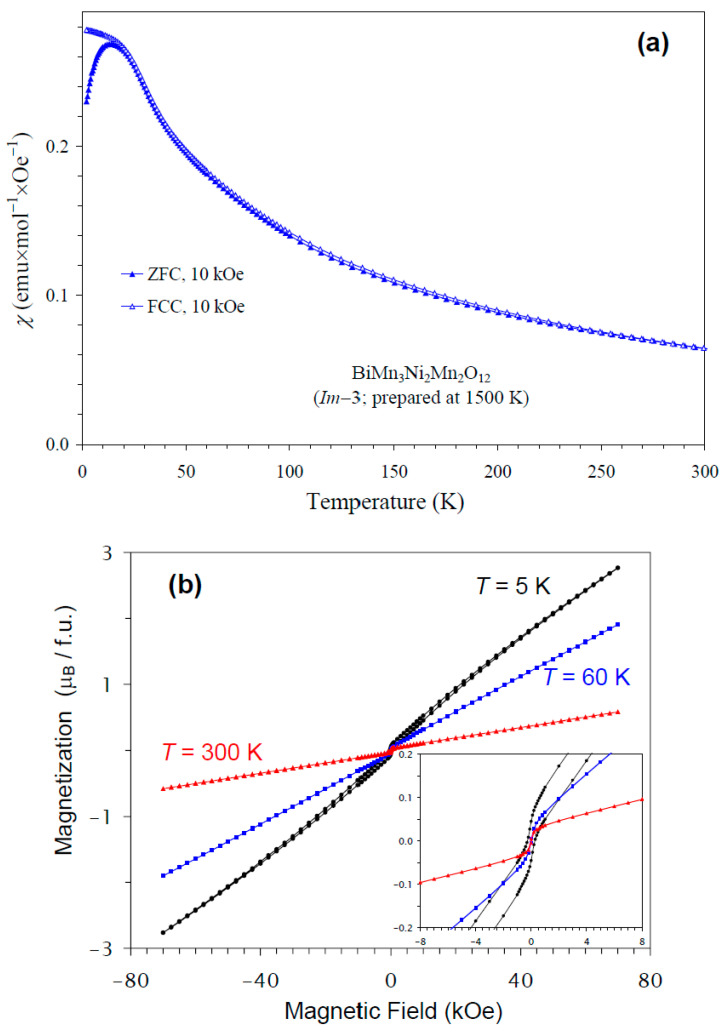
(**a**) ZFC (filled symbols) and FCC (empty symbols) dc magnetic susceptibility curves (*χ* = *M*/*H*) of BiMn_3_Ni_2_Mn_2_O_12_ measured at *H* = 10 kOe. (**b**) *M* versus *H* curves of BiMn_3_Ni_2_Mn_2_O_12_ at *T* = 5 K, 60 K, and 300 K. The inset shows magnified parts near the origin.

**Figure 6 molecules-30-01749-f006:**
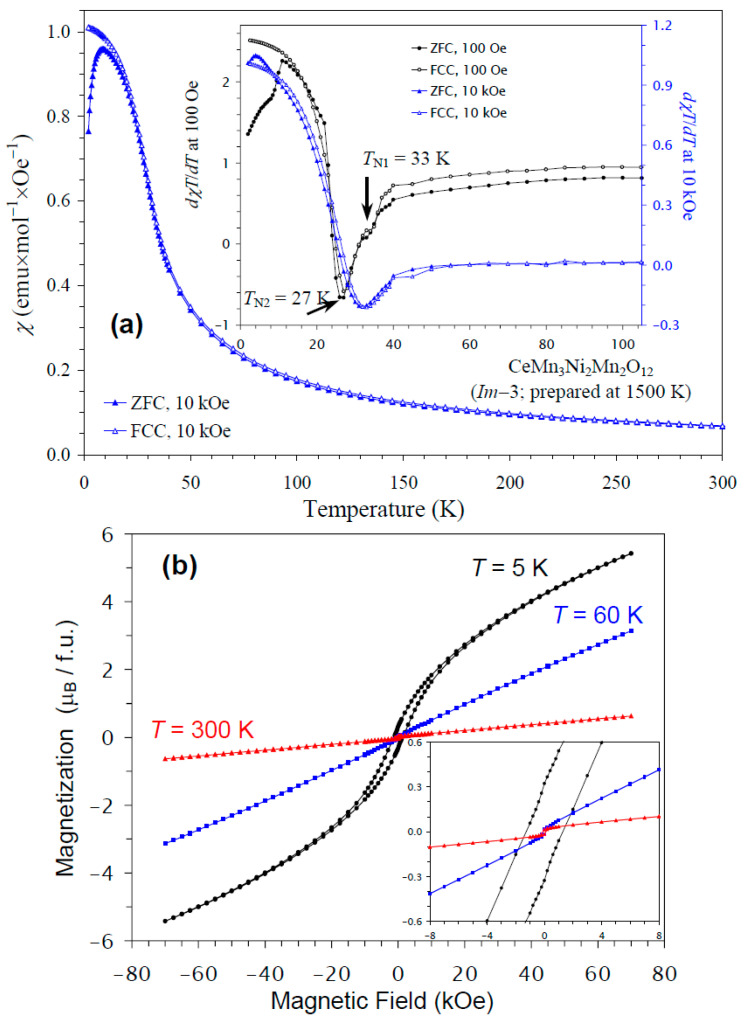
(**a**) ZFC (filled symbols) and FCC (empty symbols) dc magnetic susceptibility curves (*χ* = *M*/*H*) of CeMn_3_Ni_2_Mn_2_O_12_ measured at *H* = 10 kOe. The inset shows ZFC and FCC d*χT*/d*T* versus *T* curves at *H* = 100 Oe (black curves) and 10 kOe (blue curves). (**b**) *M* versus *H* curves of CeMn_3_Ni_2_Mn_2_O_12_ at *T* = 5 K, 60 K, and 300 K. The inset shows magnified parts near the origin.

**Figure 7 molecules-30-01749-f007:**
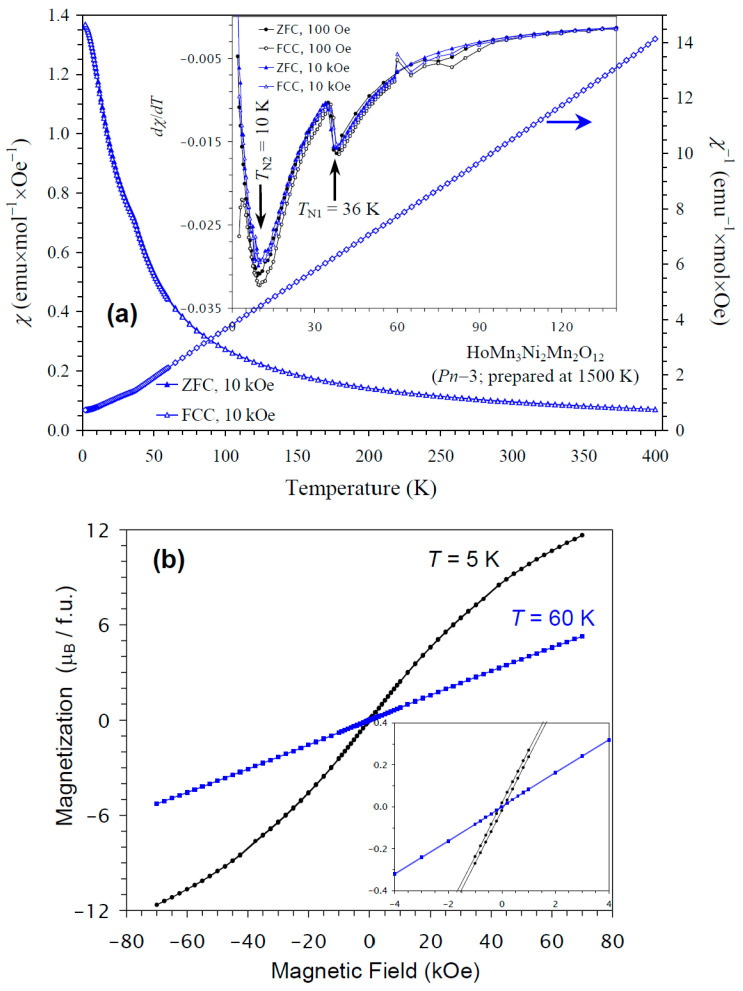
(**a**) ZFC (filled symbols) and FCC (empty symbols) dc magnetic susceptibility curves (*χ* = *M*/*H*) of HoMn_3_Ni_2_Mn_2_O_12_ measured at *H* = 10 kOe. The inset shows ZFC and FCC d*χ*/d*T* versus *T* curves at *H* = 100 Oe (black curves) and 10 kOe (blue curves). The right-hand axis shows the FCC *χ*^−1^ versus *T* curve. The arrows show the magnetic transition temperatures. (**b**) *M* versus *H* curves of HoMn_3_Ni_2_Mn_2_O_12_ at *T* = 5 K and 60 K. The inset shows magnified parts near the origin.

**Figure 8 molecules-30-01749-f008:**
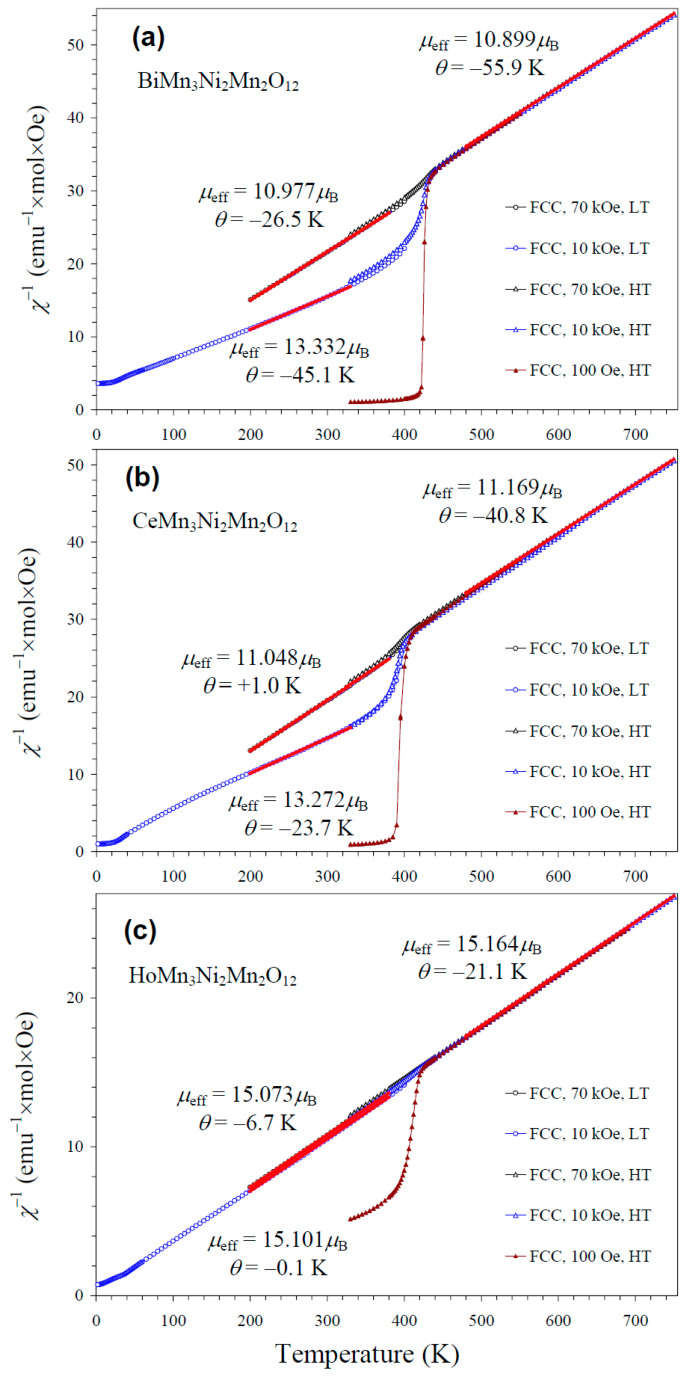
Inverse dc magnetic susceptibility curves (*χ*^−1^ = *H*/*M*) of (**a**) BiMn_3_Ni_2_Mn_2_O_12_, (**b**) CeMn_3_Ni_2_Mn_2_O_12_, and (**c**) HoMn_3_Ni_2_Mn_2_O_12_ measured at *H* = 100 Oe (brown), 10 kOe (blue), and 70 kOe (black) on cooling (FCC curves) in the low-temperature (LT) region of 2–400 K (circles) and the high-temperature (HT) region of 330–750 K (triangles). The red lines show Curie–Weiss fits (for the 70 kOe data at HT and for the 10 kOe and 70 kOe data at LT); parameters of the fits are given in the figure.

**Table 1 molecules-30-01749-t001:** Structural parameters of HoMn_3_Ni_2_Mn_2_O_12_ (*Pn*-3; prepared at 1500 K) at room temperature from synchrotron powder X-ray diffraction data.

Site	WP	*g*	*x*	*y*	*z*	*B*_iso_ (Å^2^)
Ho	2*a*	1	0.25	0.25	0.25	0.685(6)
Mn_SQ_	6*d*	1	0.25	0.75	0.75	0.507(9)
Ni_1_/Mn_1_	4*b*	0.815(12)Ni + 0.185Mn	0	0	0	0.31(3)
Mn_2_/Ni_2_	4*c*	0.815Mn + 0.185Ni	0.5	0.5	0.5	0.29(3)
O	24*h*	1	0.2578(4)	0.4242(2)	0.5567(2)	0.56(3)

Note. Crystal system: cubic; space group *Pn*-3 (No. 201, setting 2); *Z* = 2. Source: synchrotron powder X-ray diffraction (λ = 0.6200666 Å); *d*-space range used in refinements: 0.4678–7.108 Å. *a* = 7.32452(1) Å, *V* = 392.9504(9) Å^3^. WP: Wyckoff position. *R*_wp_ = 9.24%, *R*_p_ = 6.29%, *R*_I_ = 2.48%, and *R*_F_ = 2.35% after background subtraction. *R*_wp_ = 6.34%, *R*_p_ = 4.62%, *R*_I_ = 3.16%, and *R*_F_ = 3.49% without background subtraction. Occupation factors, *g*, of Ho, Mn_SQ_, and O sites are 1. Constraints on occupation factors: *g*(Mn_1_) = *g*(Ni_2_) = 1 − *g*(Ni_1_) and *g*(Mn_2_) = *g*(Ni_1_). Impurities: NiO—2.5 wt. %, HoMn_2_O_5_—3.2 wt. %, and NiMnO_3_—0.4 wt. %.

**Table 2 molecules-30-01749-t002:** Structural parameters of RMn_3_Ni_2_Mn_2_O_12_ (*Im*-3; prepared at 1500 K) at room temperature from synchrotron powder X-ray diffraction data.

R	Bi	Ce
Wavelength (Å)	0.4137875	0.6200666
Used *d*-space range (Å)	0.3122–11.855	0.4678–7.108
*a* (Å)	7.37294(1)	7.37071(1)
*V* (Å^3^)	400.7949(11)	400.4314(7)
*B*_iso_(R) (Å^2^)	2.095(9)	0.565(9)
*B*_iso_(Mn_SQ_) (Å^2^)	0.624(12)	0.521(12)
*B*_iso_(Ni/Mn) (Å^2^)	0.436(8)	0.280(9)
*x*(O)	0.31118(22)	0.31053(20)
*y*(O)	0.17728(24)	0.17562(22)
*B*_iso_(O) (Å^2^)	0.83(4)	0.62(3)
*R*_wp_ (%)	7.54 (6.22)	8.48 (6.76)
*R*_p_ (%)	5.37 (4.56)	5.73 (4.96)
*R*_I_ (%)	3.37 (3.49)	2.87 (3.67)
*R*_F_ (%)	5.99 (6.03)	1.58 (2.59)
Impurities:		
NiO	1.9 wt. %	2.5 wt. %
Bi_2_O_2_CO_3_	1.4 wt. %	-
NiMnO_3_	2.0 wt. %	5.7 wt. %
CeO_2_	-	3.4 wt. %
GdFeO_3_-type	0.5 wt. %	-

Note. Crystal system: cubic; space group *Im*-3 (No. 204); *Z* = 2. Fractional coordinates: R: 2*a* (0, 0, 0), Mn_SQ_: 6*b* (0, 0.5, 0.5), Ni/Mn: 8*c* (0.25, 0.25, 0.25), and O: 24*g* (*x*, *y*, 0). Occupation factors, *g*, of R, Mn_SQ_, and O sites are 1. Occupation of Ni/Mn site was fixed at 0.5Ni + 0.5Mn. *R* values are given after background subtraction; *R* values in parenthesis are without background subtraction. For GdFeO_3_-type impurity in R = Bi, *a* = 5.5767 Å, *b* = 7.7491 Å, and *c* = 5.4052 Å.

**Table 3 molecules-30-01749-t003:** Temperatures of magnetic anomalies and parameters of Curie–Weiss fits and *M* versus *H* curves at *T* = 5 K for RMn_3_Ni_2_Mn_2_O_12_.

R (Symmetry)	*T*_N_ (K)	*μ*_eff_ (*μ*_B_/f.u.)	*μ*_calc_ (*μ*_B_/f.u.)	*θ* (K)	*M*_S_ (*μ*_B_/f.u.)
Bi (*Im*-3)	36	10.899(2)	10.863	−55.9(3)	2.76
Ce (*Im*-3)	27, 33	11.169(1)	11.125	−40.8(2)	5.42
Ho (*Pn*-3)	10, 36	15.164(2)	15.178	−21.1(1)	11.66

The Curie–Weiss fits were performed between 480 K and 750 K using the FCC *χ*^−1^ versus *T* data at 70 kOe. *M*_S_ is the magnetization value at *T* = 5 K and *H* = 70 kOe. *μ*_calc_ was calculated using 2.4*μ*_B_ for Ce^3+^, 10.6*μ*_B_ for Ho^3+^, 4.899*μ*_B_ for Mn^3+^, 2.828*μ*_B_ for Ni^2+^, and 3.873*μ*_B_ for Mn^4+^ [67]. *T*_N_ values were determined from peaks on the 100 Oe and 10 kOe FCC *d*(*χT*)/*dT* versus *T* or *dχ*/*dT* versus *T* curves, or from specific heat data.

## Data Availability

The raw data supporting the conclusions of this article will be made available by the author upon request.

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
