# Peer review of "A Site-Ordered Quadruple Perovskites, RMn3Ni2Mn2O12 with R = Bi, Ce, and Ho, with Different Degrees of B Site Ordering"

_molecules, 2025, doi:10.3390/molecules30081749_

Round 1
Reviewer 1 Report
Comments and Suggestions for Authors
The manuscript presents a systematic investigation of A-site-ordered quadruple perovskites RMn₃Ni₂Mn₂O₁₂ (R= Bi, Ce, Ho), focusing on structural variations and magnetic properties. The work is timely and addresses an interesting topic in perovskite chemistry, particularly the interplay between cation ordering and magnetic behavior. The synthesis under high-pressure conditions and the exploration of Ce³⁺ in this system are novel contributions. However, some areas require clarification or expansion to strengthen the conclusions.
Here are major concerns. The assertion of Ce³⁺ oxidation in CeMn₃Ni₂Mn₂O₁₂, though supported by bond-valence sum (BVS) calculations, lacks direct experimental confirmation. Given the atypical presence of Ce³⁺ in perovskites, complementary techniques such as X-ray absorption near-edge spectroscopy (XANES) or X-ray photoelectron spectroscopy (XPS) are necessary to validate this claim. Furthermore, the proposed reduction mechanism of Ce⁴⁺ during synthesis requires elaboration, as the manuscript does not clarify whether this reduction occurs through oxygen non-stoichiometry or redox interactions with other cations (e.g., Mn or Ni).
The interpretation of magnetic properties also presents ambiguities. While specific heat and susceptibility data reveal transitions at 36 K (Bi/Ce) and 10 K (Ho), the origins of these transitions—whether arising from A′-site Mn³⁺, B-site Ni²⁺/Mn⁴⁺, or their cross-interactions—remain speculative. The authors’ suggestion of competing ferromagnetic and antiferromagnetic interactions to explain the small Curie-Weiss temperatures (θ) in Ce/Ho samples is plausible but underdeveloped. A more nuanced discussion, grounded in Goodenough-Kanamori rules, could elucidate specific exchange pathways (e.g., Ni²⁺–O–Mn⁴⁺ vs. Mn³⁺–O–Mn⁴⁺) and their relative contributions to the net magnetic behavior.
The manuscript presents valuable insights into the structural and magnetic properties of A-site-ordered quadruple perovskites. With revisions addressing the above concerns, it will be suitable for publication. Major revisions are required to validate the oxidation state of Ce, clarify the magnetic transition mechanisms, and strengthen structural arguments.
Author Response
Reviewer 1.
- The assertion of Ce³⁺ oxidation in CeMn₃Ni₂Mn₂O₁₂, though supported by bond-valence sum (BVS) calculations, lacks direct experimental confirmation. Given the atypical presence of Ce³⁺ in perovskites, complementary techniques such as X-ray absorption near-edge spectroscopy (XANES) or X-ray photoelectron spectroscopy (XPS) are necessary to validate this claim.
Our reply.
We thank the reviewer for this suggestion. However, we cannot perform additional experiments within the time frame provided for the revision. Moreover, it may be difficult to do such additional experiments even within few months in the future because our group does not have equipment for XANES and XPS measurements, and we will have to look for collaboration with other researchers to do such measurements (and there are possibilities that it will be impossible to find such collaborators if they are not interested in). We are asking the reviewer for understanding of this situation. We also believe that our paper has enough volume of experimental results for one independent publication.
The reviewer’ comment suggests that the main reason for the requested, additional XANES and XPS measurements is to confirm the oxidation state of Ce. In addition to the evidence from the structural analysis using state-of-the-art synchrotron X-ray diffraction, we also have additional evidence from magnetic measurements. It was not emphasized in the first version. Therefore, we added the following to the revised version: “The experimental effective magnetic moments (μeff) were very close to the calculated values (μcalc) supporting the charges of magnetic cations. In particular, the inclusion of the magnetic moment of Ce3+ [67] into the calculation gave a better agreement between μeff and μcalc; this fact gives an indirect support for the +3 oxidation state of Ce in addition to the structural data.”
We also would like to emphasize that XPS measurements are surface-sensitive and examine oxidation states in very narrow regions near the surfaces. Ce3+ can be oxidized (by air) in the surface region. For example, green (in color) MnO can become black (in color) with time (at ambient conditions) even though XRD measurements show no change in the phase composition; and the change in color suggests that the surface region is oxidized. Moreover, the sensitivity of the XPS method in determination of a ratio of different oxidation states of the same cation is very low as it depends on the fitting process of highly overlapped peaks with complications from different satellite peaks, and so on. On the other hand, magnetic measurements probe the bulk properties.
- Furthermore, the proposed reduction mechanism of Ce⁴⁺ during synthesis requires elaboration, as the manuscript does not clarify whether this reduction occurs through oxygen non-stoichiometry or redox interactions with other cations (e.g., Mn or Ni).
Our reply.
In the revised version, we provided the following explanation “The reduction of Ce4+ occurs through the oxidation of a part of Mn3+ as the high-pressure synthesis is performed in sealed capsules in closed environment.”
- The interpretation of magnetic properties also presents ambiguities. While specific heat and susceptibility data reveal transitions at 36 K (Bi/Ce) and 10 K (Ho), the origins of these transitions—whether arising from A′-site Mn³⁺, B-site Ni²⁺/Mn⁴⁺, or their cross-interactions—remain speculative. The authors’ suggestion of competing ferromagnetic and antiferromagnetic interactions to explain the small Curie-Weiss temperatures (θ) in Ce/Ho samples is plausible but underdeveloped. A more nuanced discussion, grounded in Goodenough-Kanamori rules, could elucidate specific exchange pathways (e.g., Ni²⁺–O–Mn⁴⁺ vs. Mn³⁺–O–Mn⁴⁺) and their relative contributions to the net magnetic behavior.
Our reply.
We thank the reviewer for this suggestion. According to the Goodenough-Kanamori rules, the Ni²⁺–O–Mn⁴⁺ pair should indeed have the ferromagnetic interaction. However, the Goodenough-Kanamori rules are only valid when the bond angles are close to 180 deg. As we emphasized in the introduction, RMn3Ni2Mn2O12 compounds fall into a region, where the Ni²⁺–O–Mn⁴⁺ pair can have the antiferromagnetic interaction similar to In2NiMnO6 and Sc2NiMnO6, and simple empirical rules are not valid anymore. As the second reviewer emphasized, the nature of magnetic interactions can only be understood experimentally through the magnetic structure determination with neutron diffraction. However, “those are beyond the scope of this work” (the second reviewer). Therefore, we would like to refrain from any speculative discussion at this point. For example, the magnetic structures of LaMn3Ni2Mn2O12 were determined in Ref. 12, and they are completely far from expected ones.
Reviewer 2 Report
Comments and Suggestions for Authors
In this paper high pressure-high temperature conditions have been used to synthesize three new complex perovskite compositions: AMn3Ni2Mn2O12 (A = Ce, Ho, Bi). The crystal structures are characterized with synchrotron X-ray powder diffraction. The magnetic phase transitions are characterized with SQUID magnetometry and specific heat measurements. All of the analysis looks to be correct and the behavior is similar to an earlier study with different rare-earth cations on the A-site (reference 48). One of the three compounds (A = Ho) shows partial rock-salt order of the Ni2+ and Mn4+ cations on the octahedral site. It’s interesting that this only occurs in one of the three compounds (A = Ho) and in that compound only is there a magnetic phase transition that occurs near 10 K.
Given the similarity to the compounds reported in reference 48 the impact of this paper will likely be modest. Nonetheless, there are some new twists here that I found interesting. I’m quite curious about the magnetic structures of these compounds, but to reveal the magnetic structures neutron diffraction measurements would be necessary. I concede those are beyond the scope of this work, but I hope the author will carry those out in the future. In my opinion, the overall quality and reliability of this paper makes it suitable for publication in Molecules.
I spotted a few small errors that I detail below. Those should be easily addressed in a revised version of the paper.
In the caption for Figure 1, part (a) is used where it should be (b).
On line 263 of page 10 the sentence that ends as follows, “ … at a 0.006° interval in 2θ ” Is not complete. The ending of the sentence appears to be missing.
Author Response
Reviewer 2.
- In the caption for Figure 1, part (a) is used where it should be (b). On line 263 of page 10 the sentence that ends as follows, “ … at a 0.006° interval in 2θ ” Is not complete. The ending of the sentence appears to be missing.
Our reply.
We thank the reviewer for spotting these misprints. They were corrected in the revised version.